# Youth Athletes’ Perception of Existence and Prevalence of Aggression and Interpersonal Violence and Their Forms in Serbia

**DOI:** 10.3390/ijerph19031479

**Published:** 2022-01-28

**Authors:** Radenko M. Matić, Ivana M. Milovanović, Brigita Banjac, Biljana Milošević Šošo, Jovan Vuković, Ambra Gentile, Patrik Drid

**Affiliations:** 1Faculty of Sport and Physical Education, University of Novi Sad, 21000 Novi Sad, Serbia; radenkomatic@uns.ac.rs (R.M.M.); brigita.banjac@gmail.com (B.B.); jovanvukovic89@gmail.com (J.V.); patrikdrid@gmail.com (P.D.); 2Faculty of Philosophy, University of East Sarajevo, 71420 Pale, Bosnia and Herzegovina; milosevic_biljana@yahoo.com; 3Department of Psychology, Educational Science and Human Movement, University of Palermo, 90128 Palermo, Italy; ambra.gentile91@gmail.com

**Keywords:** interpersonal violence, aggression, athletes, youth sport, Serbia

## Abstract

Aggression and interpersonal violence (IV) against children and youth are issues in contemporary society. The current study reports on the youth athletes’ perception of the aggression and IV prevalence and its manifestation forms in a Serbian context. The sample included athletes (*N* = 2091) between the age of 11 and 18 years. Data were collected through an ad-hoc questionnaire created by the authors. Respondents’ answers to introductory questions about the frequency of IV indicated its absence (78.1%). However, the answers to the questions about specific forms of violent peer behavior indicated forms as well as the time and place where IV most often occurs. They underlined that IV takes place mainly after training or competition and during sports camps; and that the dressing room is the most favorable place for these behaviors. They also confirmed that the most prevalent subtypes of IV are psychological (roughly 40%) and physical (approximately 30%). The respondents’ opinions and experiences about IV (psychological, physical, sexual) revealed that factors such as gender, location, and parents’ education level may influence the IV manifestation. Highlighting the prevalence and the most frequently manifesting types of aggression and IV could help in awareness-raising of these social issues.

## 1. Introduction

The magnitude of aggression and violence against children is a growing concern in contemporary society. There are numerous daily life situations and ways in which they (may) experience those behaviors: observing (as a witness), expressing (as a perpetrator), or being the target of it (as a victim). Although aggression and violence are inherent in all societies and different areas of social life, their presence in children’s and youth sports is particularly worrying, especially since it can often be found in the scientific literature that sport has a strong socializing character and has a positive effect on psycho-physical child development [1,2,3,4,5,6,7,8]. Therefore, it is important for young athletes to be able to recognize and define these behaviors in sport.

Researchers have established various theories on the reasons for aggressive behavior. Bandura [9] argued that children learn these behaviors by modeling (observing others) or in corroboration with reward or punishment, as he wrote in social learning theory. Another possible explanation is the theory of moral reasoning, which explains that the moral trial is an essential factor in explaining and predicting these behaviors. To observe this in another way, the theory of scenery describes how children, by observing violence in the mass media, learn aggressive scenarios, which consequently defines their direct behavior. Media has an important role in disseminating information on aggression and violence, which has a certain impact in the sports domain. Although the media provide the most prominent attention to hooliganism and vandalism, there are other minor forms of aggressive and violent behavior that should be described and explained. Furthermore, concerning the sports context, the theory of social exchange can be relevant. This is based on the premise that athletes consider the benefits and harms of their actions; they measure the punishment or damage in relation to the benefit that can be achieved. If the impermissible aggressive act is not punished enough in relation to the potential reward, the athlete will react aggressively [10]. Besides media, video games can also increase aggression in children [11]. Moreover, if the individual belongs to one group, he or she can change his/her personal identity for the social one, in which the person will behave following the group values, rules, traits, and beliefs, as described in the theory of social identity [10]. In addition to the above-mentioned theories, it is essential to clarify the distinction between aggression and violence. It is believed that there are over 200 definitions of aggression and that in most of them, we come across the key words “intent” and “causing harm” to other persons, animals, and objects [12]. Sublimating the essence of these definitions, “aggression is a behavior of a person that has an intent to harm another person, who is motivated to not become a victim of it” [13]. However, from a reductionists’ point of view (especially in the field of legal sciences, legal regulations, and practical policies, violence is defined as a form of aggression with larger intensity and goal to cause severe serious physical harm [13,14]. Consequently, all violent acts count as aggressive, but conversely, it is not the case. Based on the World Health Organization’s definition (WHO), violence is defined as “the intentional use of physical force or power, threatened or actual, against oneself, another person, or against a group or community, that either results in or has a high likelihood of resulting in injury, death, psychological harm, maldevelopment or deprivation” [15]. However, it should be emphasized that in social sciences, defining violence is a very complex task, especially since the definitions are directly conditioned by which scientific discipline researchers belong to and for what purposes they define these terms. This is also the reason for the objective differences in the scientific generation of knowledge about violence. Although we have used the (reductionist) definition of violence prescribed by the WHO for the purposes of this paper, it should be emphasized that researchers in the field of social sciences have made significant progress in understanding violence exclusively as physical and direct. Among many definitions of violence, understood in a broader sense, it is worth mentioning the definition of J. Galtung (1969), which continues to be the starting point of numerous studies on violence. Namely, J. Galtung rejects the understanding of violence exclusively as a somatic incapacitation of others. He states that “violence is present when human beings are being influenced so that their actual somatic and mental realizations are below their potential realizations” [16]. Since in this paper we point out the problem of direct/interpersonal violence, there is no room to review other relevant scientific approaches to the problem of violence. It should also be pointed out that this field research was conducted in Serbian society, which in the last decade of the 20th and the first decade of the 21st century went through a process of post-socialist and post-war transformation. Such intense cumulative social changes have left deep marks on all social actors (and on all generational groups). Hence, the current research is a research report determining the youth athletes’ perception of the presence and manifestation forms of aggression and IV in youth sport, which we understood as a specific social subsystem. Since it is a research report, we do not have specific hypotheses on the occurrence of aggressive and violent behavior in a youth sport context.

### Physical, Psychological, and Sexual Violence in the Context of Youth Sport

Aggression and violence including diverse perspectives can be categorized in numerous ways such as direct or indirect, hostile or instrumental, overt and covert, self-directed, interpersonal, and collective. Furthermore, aggression and violence can be distinguished as psychological, physical, sexual, and neglect [13,15]. The current study focuses on direct/interpersonal violence (IV), which is in line with the earlier research conducted in Serbia [17,18,19].

Non-accidental aggression and violence can take many forms such as criticism, discrimination, being shouted or sworn at, embarrassed, humiliated, teased, pushed, kicked, manipulated, gossiped, etc. [20,21,22]. There are numerous problems connected with these destructive behaviors. As a result, children can suffer potential negative and harmful consequences on their health, development, and well-being [23,24,25,26]. For instance, the child can turn in and isolate oneself, lose self-confidence, or even burn out and leave the sport. Young people tend to have a positive picture of playing sports [20]. Even though children had experienced this negative behavior, their value of participation in sport outweighed their discomfort, and if they could choose, they would participate again in physical activity [26]. On the other hand, children can participate less in team sport after they suffer from unwanted violence and maltreatment [27].

More profound knowledge on the factors that elevate the risk of aggression and violence victimization is essential to create more effective prevention initiatives toward aggressive and violent behaviors. Based on the scientific literature, those factors can be age, gender, sport type, ethnicity (minority group), athletic ability (low), economic state (poor), sport level (elite), number of hours of weekly practice, disabled, and LGBT athletes [22,27,28,29]. Hence, children and adolescents who can be characterized by some of these factors are primary in terms of vulnerability.

Many large-scale studies have investigated this topic in the field of youth sports and demonstrated its significance. In 2020, Parent and Vaillancourt-Morel [28] conducted a study on 1055 Canadian athletes to observe the magnitude of IV against children in the sporting context. Their findings indicated that 9.2% of the respondents had at least one experience of psychological violence, 39.9% physical violence, 35.7% neglect, and 28.2% sexual violence. Similarly, Vertommen et al. [29] examined the prevalence of IV in sport in the Netherlands and Belgium by surveying 4000 adults about their experiences with childhood violence while playing sport. As a result, 38% had an experience with psychological, 11% with physical and 14% with sexual violence before the age of 18. Another study among more than 6000 adults from the UK reported some negative or harmful experiences in sport as children in the following form: 75% emotional harm, 29% sexual harassment, 24% physical, and 3% sexual harm [20].

Much of the empirical evidence confirms that the most prevalent aggression form against children and adolescents is verbal [21], especially among girls [28,29]. Generally, this form has the highest attention among peers. Interestingly, girls’ verbal aggression is severe and hurts more than when the boys do it [30]. In contrast to these findings, boys are dominantly the perpetrators when we look at the presence of physical violence [18]. Additionally, in most studies, boys reported a higher proportion of physical violence experience than girls [20,28,29]. In the past, the first association with the sexual violence term was related to the coach–athlete relationship. The recent tragedy in elite USA gymnastics is one direct example of this, where the ‘army of the survivors’ (156 women athletes) spoke about their experiences as victims. In short, their doctor Larry Nassar sexually abused them for up to 10 years in terms of ‘medical treatment’ [23]. We also found studies with contradictory results for this kind of violence, where one study found no differences in terms of gender about the occurrence of sexual violence [28], while another highlighted that girls had a higher experience rate [29]. It is also important to note that athletes can also be perpetrators as well as victims of any kind of violence [20,24,30].

Addressing these unsportsmanlike behaviors in the sporting context is relevant because athletes of all ages and competitive levels are exposed to it [24]. Therefore, education is essential to build and disseminate awareness on non-accidental aggression and violence [23,27,30], even if it is a sensitive area. The children’s social environment, namely the local community, school, sports clubs, and the family, have the power to prevent, recognize, and manage these issues. Thus, identifying prevention programs, strategies, initiatives, or training courses would be effective in the fight against aggression, violence, and exclusion episodes [22,25] such as the “Sport against Violence and Exclusion” (SAVE) project [31].

Summarizing the results of previous research, we can state that aggression and IV exist in organized youth sport. However, previous research has mainly focused on examining the manifestations of aggressive behavior and IV in sport, neglecting some determinants such as time and place of aggression and IV manifestation. Accordingly, it seems that sport theory and practice lack information and the timing of aggressive behavior and violence, with the aim of better social intervention. This research intends to examine the perception of youth athletes of the existence and manifestation forms of aggressive behavior and IV in sport and its frequency, with an analysis of the place and time of its occurrence. The analysis of the status of sociodemographic characteristics of the respondents will also contribute to a better understanding of the researched determinants of aggressive behavior and IV.

## 2. Materials and Methods

### 2.1. Participants

The study involved children and youth (*N* = 2091) from the territory of the Autonomous Province of Vojvodina. Their ages ranged between 11 and 18 years, attending primary or secondary/high school. Furthermore, they participated regularly in one or more of the following sports: archery, athletics, body shaping sports, combat sports, cycling, dance, equestrian sports, gymnastics, ice sports, nautical sports, racquet sports, small ball sports, team sports, water sports, and yoga.

All sampling processes of reducing the participant from a total sample size of children to the sample used in the present study are shown in Figure 1.

Thus, among the total and eligible sample (N = 3000), 763 were excluded because they did not deliver the written informed consent. Furthermore, among the participants who accepted written informed consent (N = 2237; 74.56%), data from 146 participants were excluded due to missing data at least one variable of age, location, father’s education, mother’s education, number of children, sport, is there any violence in the sport you practice, and how often does violence in your sport occur among children.

Based on the relevant information from the Statistical Office of the Republic of Serbia, the total population of children from the territory of Vojvodina from 11 to 18 years is 148,493, while there are 3023 sports organizations. Therefore, the final sample in this study included 2091 participants, with a response rate of 69.7%, representing 1.4% of the total population of the Autonomous Province of Vojvodina.

The research was approved by the Faculty of Sport and Physical Education Ethics committee with the official number: 46-11-07/2020-1.

### 2.2. Measurements

This research was carried out using a questionnaire designed to collect data on the topic of youth athletes’ perception on IV existence and manifestation forms in youth sport. This method was chosen because of its nature to serve research with a very clearly defined problem and research objectives, with strict sampling designs and sophisticated analysis of the results. The benefits of applying this method for the purposes of this research are reflected in the ability to determine the behavior of respondents based on their attitudes, abilities, opinions, characters, and emotions while meeting the principles of rationality and economy. In the Serbian educational system (within the subjects Civic Education, Psychology, Sociology), children aged 11 to 18 mostly learn about violence (peer violence, adult violence against children etc.). Hence, in order to achieve terminological clarity in the questionnaire, respondents were asked questions about violence and not about aggression, in order to avoid possible misunderstanding of the questions. The survey was composed of the following sections:(1)Description of the respondents’ socio-demographic characteristics: gender, age, location, father’s education, mother’s education, number of children in the family, and sport.(2)Questions for perception of IV presence which included: (a) frequency of the aggression and IV among athletes (from 1—never to 5—very often); (b) forms of IV that can occur (in theory) contain three items related to physical, psychological, and sexual violence and indicate the respondents’ perceptions of possible forms of aggressive behavior and violence that may befall them during their sporting activities; (c) forms of IV that occurred most frequently (in practice), which explains the forms that are really happening in real sporting life of the respondents (physical, psychological, sexual); (d) time; (e) location where IV occurred; and (f) forms of IV to cover the whole experience of these destructive behaviors. The time of occurrence of violence included 8 items on the timing of aggressive behavior in relation to the arrival phase, during, or after training, competition or preparation in sports (before, during, and after training, before, during, and after competition, during travel, at sports camps). On the other hand, the variable venue included 4 items, concretizing the space of occurrence of IV (gym, dressing room, training area, bathroom/toilet area).

The authors of this research compiled the questionnaire, which was included in earlier studies [17,18]. Prior to field research, a pilot study has been performed with limited number of respondents (*N* = 60). The questionnaire showed satisfactory metric measurable results. The data obtained from this pilot study generally showed good reliability and satisfactory validity of the new measuring instrument. Thus, the reliability of the scales can be expressed via the Cronbach’s alpha coefficient (α) of internal consistency from 0.81 to 0.93. The validity of the new measuring instrument was determined by linear correlation analysis (linear correlation coefficient, r), where the correlation coefficient r (validity measure) ranged between 0.74 and 0.80.

The answers were measured on a five-point Likert scale, from completely disagree (1) to completely agree (5). As already pointed out, the questionnaire consisted of questions whose answers would indicate whether respondents were aware of what manifestations of IV behavior exist as well as questions about which forms of those behaviors really exist in the sport they train in. The survey was conducted anonymously.

### 2.3. Procedures

A cross-sectional design was used for the study’s construction. The research was conducted between October 2019 and October 2020. In order to ensure the best possible treatment of research, distribution of all questionnaires was forwarded through the Association of Physical Education Teachers of Vojvodina and Association for school sports of the city of Novi Sad, which contributed to quite a good process of the collection of data. Executives from these associations organized and randomly distributed 3000 questionnaires to children who participated regularly in one or more of the above-mentioned sports in randomly selected sports clubs among a network of sports clubs in Vojvodina (3023 sports clubs). Moreover, the study involved villages, suburbs, towns, and cities, and the process of collecting the data was conducted at the introduction part of training in sports clubs. The questionnaire completion took up to 30 min including the basic instructions for the support of children and youth participants for filling in the questionnaire by the researchers and coaches. More importantly, they explained the terminology used, which could be unknown to athletes. Additionally, each participant took part in the study on a voluntary basis. Participants, their parents, and club representatives were informed about the purpose of the research with the emphasis that all data are anonymous and will be used exclusively for scientific research purposes. Finally, all data were processed and managed in accordance with the legislation on the protection of personal data and the General Data Protection Regulation (GDPR).

### 2.4. Data Analysis

First, the descriptive statistics of the socio-demographic characteristics and the occurrence of IV were calculated. Then, the Chi-squared test and the Mann–Whitney U test were deployed to examine the differences. The collected data were analyzed through IBM SPSS version 24.0. The statistical significance level was set at *p* < 0.05.

## 3. Results

### 3.1. Descriptive Characteristics of the Sample

Table 1 presents the number and frequency of the participants by gender, age, location, father’s and mother’s education, number of children in the family, and type of sport. It can be observed that the prevalent gender was male (60.9%). Besides, the largest percentage of the children and youth lived in towns or cities (78.1%) and attended primary school (63.5%). Their parents’ educational level was roughly equal; slightly dominant for the secondary/high school category. Hence, it allows for investigations of the influence of the parents’ educational level on the children’ attitudes toward aggressive behaviors such as IV. In addition, over 70% of families had up to two children. Finally, team (73.2%), combat (8.6%), and water (8.2%) sports were prevalent.

### 3.2. Attitudes toward IV

Based on the results from Table 1, in variable “Is there violence in the sport you practice”, which evaluated only the respondents’ perception of violence in the sport in which they practice, 68% of athletes did not detect any forms of aggression or violence in their sport. Thus, it is not surprising that the vast majority stated neutral or negative attitude toward those behaviors. Furthermore, in the next control variable “How often does violence in your sport occur among children”, the obtained results found almost 10% lower frequencies of IV (78.1%, summary of rarely and never categories). Therefore, responses in these two variables showed some differences between the respondents’ perceptions on existing IV in general and its frequency. Therefore, the results in these two variables from a general context needed an additional specified explanation in the context of place, time, and form of IV, which is presented in Table 2.

### 3.3. Place, Time, and Form of IV

In Table 2, responses underlined that the preferred time for IV was the period after training (20.9%) or competition (15.8%) and during sports camps (15.1%). In line with this, almost 30% of athletes identified the dressing room as the most favorable place for these negative behaviors. Another key finding is that the most prevalent form of IV that could occur and the most common type were psychological (roughly 40%) and physical (approximately 30%).

### 3.4. Associated Factors with IV

The results in Table 3 revealed the existence of gender difference in the answers concerning IV occurrence. Although both girls and boys indicated a low presence of IV in their sport, girls reported IV occurrences as less frequent than males (Boys_yes_ = 37.0%; Girls_yes_ = 24.2%; χ^2^ = 39.65, *p* < 0.05). Furthermore, there was a significant association between the number of children in the family and IV manifestation. If the family had three or more children, the above-mentioned destructive behavior was less noticeable, but it appeared more frequently than in families with up to two children (Less than 2 children_yes_ = 33.4%; More than two children_yes_ = 28.7%; χ^2^ = 4.19; *p* < 0.05).

### 3.5. Differences among Different Types of IV

The respondents’ opinions and experiences on IV (psychological, physical, sexual) in sport are presented in Table 4. A higher perception of physical (U = 521,237.5, *p* < 0.01) and psychological (U = 490,568.0, *p* < 0.01) violence was found among girls compared to boys. Furthermore, girls also indicated a higher level of psychological violence in practice (U = 525,082.0, *p* < 0.01). Interestingly, sexual violence was significantly related to villages compared to other locations, both in theory (U = 355,822.0, *p* < 0.01) and in practice (U = 347,470.0, *p* < 0.01).

Finally, significant differences were found among the parents’ educational levels. Specifically, lower educational levels of the mother and father were related to perceived physical (U_Father_ = 469,877.5, *p* < 0.01; U_Mother_ = 441,360.0; *p* < 0.01), psychological (U_Father_ = 481,308.0, *p* < 0.01; U_Mother_ = 454,255.0; *p* < 0.01), and sexual violence (U_Father_ = 475,519.5, *p* < 0.01; U_Mother_ = 436,413.0; *p* < 0.01) and actual physical (U_Father_ = 486,243.0, *p* < 0.05; U_Mother_ = 457,064.0; *p* < 0.05) and sexual (U_Father_ = 470,656.5, *p* < 0.01; U_Mother_ = 441,654.0; *p* < 0.01) violence.

## 4. Discussion

The current study reports on the youth athletes’ perception of the existence and manifestation forms of aggression and IV in youth sport in north Serbia (the Autonomous Province of Vojvodina). Specifically, apart from the above mentioned, this study considered some factors that may contribute to the occurrence of these destructive behaviors. The target population were athletes attending elementary and high school. Their answers were collected via questionnaires. Interestingly, the findings of this study indicate some contradictory claims.

Measuring the prevalence of sport-related IV among children and youth athletes is difficult since it often takes place latently. Namely, 78.1% (when we combined answers “rarely” and “never”) of athletes denied any kind of IV in their sport. The fact that their attitude was negative in the first place toward IV is notwithstanding with their additional answers. Therefore, with latent answers, the opposite was determined. A possible explanation can be that this topic is really sensitive, and in most cases, perpetrators and victims are reluctant to discuss or report these issues [21,27]. In light of this, children and youth do not feel comfortable speaking up [20] because showing bad behaviors such as IV is wrong and socially “inappropriate”. Furthermore, children and youth are sometimes not even aware that they are victims [21] because of the “normalization“ of those destructive behaviors in their social environment [20,26]. This is due to the fact that in the last few decades, children and youth in Serbian society grow in an environment that is manifestly peaceful and democratic, and latently tolerant of violence that is all-pervasive in real and virtual environment (in school, on the street, in city transport, in the media, in social networks). Growing up in such a social environment can negatively affect the perceptions of children and youth of what violence is or is not. We determined this by asking introductory as well as control questions about the respondents’ perception of whether there was IV in youth sports. It turned out that 78.1% of respondents answered negatively to this question. However, the answers to the questions with offered examples of violent behavior indicate that the respondents perceived that the existence of violent behavior was mainly psychological (almost 40%) and physical (about 30%). By analyzing the subtypes of IV in our results, the most prevalent form that could occur (in theory) and the most common (in practice) are psychological (roughly 40%) and physical (approximately 30%). Correspondingly, the scientific evidence also underlines that the psychological subtype is the most prevalent [20], followed by the physical and sexual [21,28,29]. Indeed, the psychological type is the core of all other forms [24].

Moreover, the results indicate that these non-accidental harms perpetrated on athletes are mostly typical for the period after training or competition and during sports camps, which is in line with the previous studies [25,27]. Moreover, almost one third of respondents identified the dressing room as the place where it usually happens. It is understandable because children are still under the games’ emotions, which can cause excessive reactions. It is important to mention that generally, the dressing room is the only unsupervised place from coaches and parents, within the spatial framework of this research; thus, children and youth can act according to their own rules and will, without sanction [19].

Therefore, this research report indicates the fact that sport is not immune to the wider social circumstances including aggression and IV, even though every child has a right to a safe sport that is respectful, equitable, and free from those destructive behaviors [24,27]. Looking at the factors that may contribute to the occurrence of aggression and violence, gender characteristics significantly influenced their manifestation as well as in one earlier study [26]. Although girls and boys did not indicate high occurrence of aggression and IV in the sports domain, boys were more involved in those destructive behaviors in this research. Similarly, Vveinhardt et al. [21] stated that boys tended to be more likely to be involved in bullying. Furthermore, the number of children within a family had an impact on the frequency of IV. If they had three or more children, those destructive behaviors were less noticeable, but their appearance frequency was higher.

Finally, the athletes’ perception and experiences on IV (psychological, physical, and sexual) showed that factors such as gender, location, and parents’ education level influenced the occurrence of such various forms. The first question was related to the opinion of children about the IV forms that can occur, while the second question was about their experiences, and how the types of those destructive behaviors manifested in practice. Findings suggest that based on gender, girls have a higher perception of psychological and physical violence than boys. Additionally, there were also statistical differences for physical violence (only in theory). It also underlines the fact that psychological violence is the most dominant subtype of all, especially among girls [18,28,29]. Furthermore, the location indicated that respondents living in villages or suburbs had a higher level of sexual violence compared to their peers living in towns or cities. Finally, the parents’ educational level had a significant impact. If they finished only primary or secondary education, their children characterized higher scores of all three types of violence that can occur in theory, and physical and sexual violence manifestation in practice. Moreover, athletes from families with one or two children tended to report more sport violence than children living in families with three or more children. A previous study showed that having two or more siblings predicted the occurrence of violence among siblings [32]. Similarly, athletes coming from families with three or more children may consider IV as more “common” than children living only with parents (without siblings).

### Limitations and Recommendations for Future Studies

The limitations of this research should be noted. The research process was difficult because of the emergency measures that were in force in Serbia due to the COVID-19 pandemic. The sample representativeness could be improved with larger sample size and a mainly equal number of gender and sporting types. In addition, this research was geographically limited to Serbia’s autonomous northern part, which may not provide the real picture of the occurrence of aggression and IV across the whole country. Apart from this, a longitudinal study design would be more appropriate to evaluate the aggression and violence manifestations in greater depth.

## 5. Conclusions

The research findings underline the youth athletes’ perception on the existence of aggressive and violent behavior and manifestation in organized youth sport in the Serbian context. Although the majority of the respondents answered to the introductory questions that “there is no violence” in their sport, the answers to the further questions indicated different findings. Such incompatibility (about the existence of IV in youth sports in general) and the latter questions about specific forms of IV can be understood as insufficient sensitivity of respondents to what the term “violence” means to them. This can be supported by the answers to later questions that indicated the existence of some forms of IV in the field of psychological and physical violence. Furthermore, this study analyzed some important IV determinants such as time and place. As a result, the most favorable times for IV were the periods after training or competition and during a sports camp. The dressing room was marked as the most dominant place. For better understanding of these destructive behaviors, the study revealed that some factors such as gender and the number of children in the family were associated with the occurrence of IV. Besides these aspects, the location and the parents’ education level influenced the manifestation of various violence forms (psychological, physical, and sexual) that can occur in theory and in practice.

Moreover, this field research in the sports domain has several (practical) implications. First, it showed that organized youth sport is not *a priori* good or bad for children but has the potential of producing both positive and negative outcomes. Second, bearing in mind the general social circumstances in Serbian context, it potentially indicates the insufficient sensibility of children and youth to what violence in sport is or is not. Finally, by highlighting the occurrence and the most manifesting forms of aggression and IV, these results could help athletes, coaches, parents, caregivers, etc. in raising the awareness of these problems. Although future research could define practical tools for recognizing, preventing, and reducing aggression and IV in organized youth sport, this study could provide a useful empirical basis toward that goal.

## Figures and Tables

**Figure 1 ijerph-19-01479-f001:**
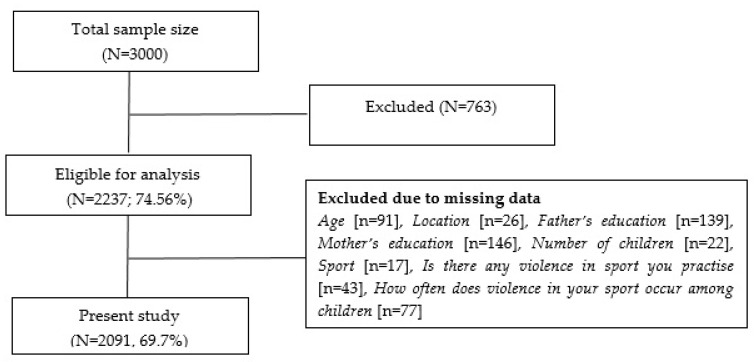
The sampling process.

**Table 1 ijerph-19-01479-t001:** Descriptive statistics for the samples’ socio-demographic characteristics and number and frequency of IV (interpersonal violence) in sport based on the participants’ answers.

Variable	*N*	%
Gender		
Female	874	39.1
Male	1363	60.9
Age		
Primary school (11–14 years of age)	1363	63.5
Secondary school (15–18 years of age)	783	36.5
Location		
Village or Suburb	484	21.9
Town or City	1727	78.1
Father’s education		
Primary or secondary education	1000	47.6
College or higher education	1098	52.3
Mother’s education		
Primary or secondary education	809	38.7
College or higher education	1282	61.3
Number of children in the family		
1 or 2 children	1659	74.9
3 or more children	556	25.1
Sport		
Archery	2	0.1
Athletics	34	1.5
Body shaping sports	6	0.1
Combat sports	192	8.6
Cycling	6	0.3
Dance	78	3.5
Equestrian sports	1	0.0
Gymnastics	48	2.1
Ice sports	8	0.3
Nautical sports	9	0.4
Racquet sports	11	0.5
Small ball sports	2	0.0
Team sports	1638	73.2
Water sports	184	8.2
Yoga	1	0.0
Is there violence in the sport you practice?		
Yes	702	32.0
No	1492	68.0
How often does violence in your sport occur among children?		
Very often	45	2.1
Quite often	59	2.7
Sometimes	370	17.1
Rarely	596	27.6
Never	1090	50.5

**Table 2 ijerph-19-01479-t002:** Descriptive statistics of IV relying on the respondents (%).

Variable	I Completely Disagree	I Disagree	I Am Neutral	I Agree	I Completely Agree
When does violence among children occur?					
Before training	46.9	21.9	16.9	11.1	3.2
During training	47.3	23.3	16.7	9.8	2.9
After training	42.2	20.4	16.6	17.0	3.9
Before competition	51.9	24.7	15.5	5.4	2.4
During competition	47.5	20.6	17.7	10.5	3.6
After competition	45.5	20.4	18.3	11.7	4.1
While travelling to competitions, sports camps, etc.	48.7	21.0	17.2	9.2	3.9
In sports camps	46.5	18.1	20.3	10.9	4.2
Where does violence among children occur?					
In the gym	48.1	24.3	16.1	9.5	2.0
In the dressing room	38.1	16.6	16.0	22.3	7.0
In the training area	50.4	24.7	16.1	6.4	2.4
In the bathroom/toilet area	54.8	19.8	14.0	7.9	3.9
Forms of violence that can occur (in theory)					
Physical violence	28.8	17.3	20.0	22.5	11.5
Psychological violence	21.7	15.6	18.9	28.2	15.6
Sexual violence	62.1	16.1	13.8	4.1	3.7
Most common forms of violence that have occurred (in practice)					
Physical violence	32.6	16.9	19.6	19.6	11.3
Psychological violence	27.0	13.6	18.6	24.0	16.8
Sexual violence	69.5	14.0	12.4	1.9	2.2

**Table 3 ijerph-19-01479-t003:** Results of Chi-squared test among variables of the occurrence of IV.

Variables	Is There Violence in the Sport You Practice? (%)		How Often Does Violence in Your Sport Occur among Children? (%)				
	Yes	No	Very often	Quite often	Quite often	Rarely	Almost never
Boys	37.0	63	2.8	3.0	20.0	30.7	43.4
Girls	24.2	75.8	0.9	2.2	12.2	22.7	61.5
	χ^2^ = 39.65 *		χ^2^ = 71.60 *				
Village or Suburb	28.4	71.6	1.1	1.9	15.5	28.7	52.8
Town or City	32.8	67.2	2.3	3.0	17.6	27.2	49.9
	χ^2^ = 3.28		χ^2^ = 6.08				
Primary or secondary education (Father)	30.9	69.1	1.7	3.1	16.9	25.8	52.4
College or higher education (Father)	33.5	66.5	2.2	2.4	17.6	29.8	48.0
	χ^2^ = 1.55		χ^2^ = 6.28				
Primary or secondary education (Mother)	32.6	67.4	1.9	3.6	17.3	26.3	51.0
College or higher education (Mother)	32.4	67.6	2.0	2.2	17.5	28.8	49.5
	χ^2^ = 0.01		χ^2^ = 4.62				
Family up to two children	33.4	66.6	1.9	2.2	17.5	27.5	50.8
Family with three and more children	28.7	71.3	2.9	4.4	15.8	28.1	48.9
	χ^2^ = 4.19 *		χ^2^ = 9.53 *				

Legend: * *p* < 0.05.

**Table 4 ijerph-19-01479-t004:** Results of the Mann–Whitney U test among variables in the forms of IV.

Variables	Forms of Violence that Can Occur (in Theory) (Mean Rank)			Forms of Violence that Occurred Most Frequently (in Practice) (Mean Rank)		
	Physical violence	Psychological violence	Sexual violence	Physical violence	Psychological violence	Sexual violence
Boys	1058.44	1037.05	1079.29	1093.48	1061.53	1085.45
Girls	1138.69 **	1183.26 **	1100.46	1060.06	1127.66 **	1055.93
Village or Suburb	1108.77	1067.44	1140.97 **	1068.13	1037.31	1113.80 **
Town or City	1069.13	1085.26	1057.92	1068.60	1084.30	1048.46
Primary or secondary education (Father)	1073.08 **	1069.28 **	1063.22 **	1039.97 *	1023.98	1044.18 **
College or higher education (Father)	975.28	986.33	980.26	989.71	1015.50	974.35
Primary or secondary education (Mother)	1079.88 **	1072.09 **	1082.00 **	1042.98 *	1034.44	1050.92 **
College or higher education (Mother)	979.46	990.81	974.85	990.33	1006.24	976.46
Family up to two children	1066.71	1081.32	1074.80	1067.09	1071.76	1061.19
Family with three and more children	1118.21	1090.09	1085.65	1080.83	1092.77	1072.54

Legend: * *p* < 0.05, ** *p* < 0.01.

## Data Availability

The raw data supporting the conclusions of this article will be made available by the corresponding author without undue reservation.

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
