# Peer review of "Youth Athletes’ Perception of Existence and Prevalence of Aggression and Interpersonal Violence and Their Forms in Serbia"

_ijerph, 2022, doi:10.3390/ijerph19031479_

Round 1

Reviewer 1 Report

From what I have seen, this article extends previous studies on that subject in Serbia. The sample has a considerable number of participants, which helps to consider the results.

Furthermore, most of the variables are relevant to understanding this critical phenomenon (time, place, the influence of gender, location, parental education).

On the other hand, the authors have repeated two unclear questions in their study repeatedly with the same contradictory answers. It seems that children do not have specific forms of violence in mind because the questions "is there violence in the sport you play?" and "How often does violence occur in your sport among children?" (Table 1) refer to violence non-specifically.

Suppose the researchers want to get different answers. In that case, they should ask different questions that lead the children to specific memories or observances among the different types of violence the study considers (pushing, hitting, forcing to do things that annoy me; insult, threaten; isolate, humiliate, touch).

Authors should consider that option among the limitations and conduct further studies to see the differences. Reconsider discussion as well.

Line 31: This sentence is not clear: “Therefore, it is inevitable for athletes not to encounter these deviant, latent behaviours in sport”.

Line 56: the violence definition and the theoretical framework. They can be improved at both a structural and cultural level (for example, according to Galtung).

Line 150: "The children were asked about violence and not about aggression." This information is inconsistent with the title and on line 156, "questions for aggression and IV presence".

Line 197: the expression "dominant gender" should be changed to a neutral one.

Lines 266 and 267: quotation marks are irregular in words "inappropriate" and "normalization".

Author Response

Dear Reviewer,

thank you for the constructive criticism and suggestions.
We have included them in the manuscript. All new text in the manuscript is colored, which is why you will easily see that we have made changes and additions.

Reviewer 2 Report

  1. This article's title is not reflective of the study findings. The appropriate title and focus should be " Youth Athletes' Perception of the prevalence of Aggression and Interpersonal Violence and their Forms in Serbia".
  2. The content in the abstract should also by tailored to reflect the findings where 78.1 percent of the respondents perceive that there is no interpersonal violence in Serbian sport among the youth athletes. It is intellectually dishonest to focus on the perception of  the minority of respondents and ignore or explain away the predominant responses which are that there is no major problem regarding Aggression and IV. In short, the thrust of the abstract is misleading.
  3. The word attend on page 1 line 30 can be replaced with "experience" and the sentence edited for clarity.
  4. The tense for line 38 - 40 should be checked for suitability.
  5. On page 2, lines 52 to 56, are not clear and require editing for clarity. Similarly the sentence on WHO's definition of 'violence'.
  6. Page 2, line 75, the tense and grammar need revisiting.
  7. Line 84, page 2, the word 'which" should be replaced with 'who'.
  8. Page 3. lines 101 (replace that with these)....but also rephrase sentence to make sense. On line 109, replace 'other' with 'another'.
  9. Page 3, line 119, SAVE, sounds like it is an abbreviation.....should therefore be  written in full. Lines 122 and 123 need rephrasing for clarity and also lines 130 - 134. The aim of the study should be articulated better.....the focus is on perception of prevalence and forms!
  10. On page 3 under 'Materials and Methods', the authors should spell out how the sampling was done. Also regarding the methodology, it should be explained whether the questionnaire items were revised after the piloting. The characteristics of the 60 pilot participants should be highlighted. Additionally, the authors don't clarify whether they received IRB approval and how they dealt with ethical considerations before collecting data.  Additionally, the procedure of administering the questionnaire and venues are not clear. These are core issues for the methodology that need addressing.
  11. Page 4, line 200, 'second' sounds like it should be 'secondary'.
  12. Results are very clear that 78.1 percent of the respondents perceive that aggression and IV do not exist.....however, the authors downplay this by saying on page 6, line 208 that "almost 70% did not"! And on line 212, the authors assert that "one third of respondents underlined (with their neutral and positive answers"! How can neutral be lumped together with positive? It is clear that only 20.9 percent (from table 2) indicate that IV occurs after training. The skewed reporting of results does not do justice to the data collected and shows a total lack of objectivity by the researchers.
  13. Some honest reporting and interpretation of the results is needed. Given the lack of objective interpretation of the data, the section on discussion is also compromised. The assertions on page 9 are skewed as they are in line with the skewed results reported. This continues into the conclusion where the overarching finding that the majority (78.1 percent) of the respondents perceive that there is no prevalence of aggression and IV in Serbian Youth Sport is totally ignored.   
  14. The authors need to revisit this paper and address the above concerns with due honesty and objectivity. 

Author Response

Dear Reviewer,

thank you for the constructive criticism and suggestions.
We  included them in the manuscript. All new text in the manuscript is colored, which is why you will easily see that we have made changes.

Round 2

Reviewer 2 Report

Good job on the revision. Please, do grammatical verification throughout the manuscript. Otherwise, a big upgrade from previous submission.